# Birdshot Chorioretinopathy: A Review

**DOI:** 10.3390/jcm11164772

**Published:** 2022-08-16

**Authors:** Elodie Bousquet, Pierre Duraffour, Louis Debillon, Swathi Somisetty, Dominique Monnet, Antoine P. Brézin

**Affiliations:** 1Department of Ophthalmology, Ophtalmopôle, Hôpital Cochin, Assistance Publique-Hôpitaux de Paris, AP-HP, Université Paris Cité, 75014 Paris, France; 2Retinal Disorders and Ophthalmic Genetics Division, Stein Eye Institute, University of California Los Angeles, Los Angeles, CA 90095, USA

**Keywords:** birdshot chorioretinopathy, multimodal imaging, HLA-A-29, posterior uveitis, immunosuppressive treatment

## Abstract

Birdshot chorioretinopathy (BSCR) is a bilateral chronic inflammation of the eye with no extraocular manifestations. BSCR affects middle-aged individuals from European descent and is strongly associated with the human leucocyte antigen (HLA)-A29 allele. The immune mechanisms involved are not fully understood, but recent advances have shown the role of Endoplasmic Reticulum Aminopeptidase 2 (ERAP2) in disease pathogenesis. Multimodal imaging, including fluorescein angiography, indocyanine angiography, fundus autofluorescence, and optical coherence tomography, are useful in confirming the diagnosis and monitoring disease activity. Visual field testing is also important to assess the disease progression. To date, there is no consensus for optimal treatment regimen and duration. Local and systemic corticosteroids can be used for short periods, but immunosuppressive or biological therapies are usually needed for the long-term management of the disease. Here, we will review publications focused on birdshot chorioretinopathy to give an update on the pathophysiology, the multimodal imaging, and the treatment of the disease.

## 1. Introduction

Birdshot chorioretinopathy (BSCR) is presumably an auto-immune disease of the eye, affecting middle-aged individuals from European descent [1]. Ryan and Maumenee introduced the term “birdshot chorioretinopathy” in 1980, describing the cases of 13 patients with multiple white-creamy choroidal spots that appear like a birdshot from a shotgun [2]. The following year, Gass described additional cases, which he called “vitiliginous chorioretinitis” due to the appearance of the choroidal lesions [3].

The exact pathogenesis remains poorly understood, but the disease is strongly associated with the human leucocyte antigen (HLA)-A29 allele [4]. Histologic analyses of eyes with BSCR revealed non-granulomatous nodular infiltration of the choroid and lymphocytic infiltrates [5,6]. The disease is strictly localized to the eye with no extraocular manifestations or systemic disease associations. 

BSCR manifests as a chronic, bilateral, posterior uveitis characterized by multiple white-creamy choroidal spots [4]. The range of BSCR phenotypes varies between a benign form of the disease which has a good prognosis, to sight-threatening, progressive chorioretinal atrophy. BSCR is usually treated by corticosteroids, immunosuppressants, or immunomodulating biological therapies.

The aim of this review is to provide an update on the diagnosis with a focus on chorioretinal imaging techniques to assess the disease activity and monitor the disease progression.

## 2. Epidemiology

BSCR is a rare chronic posterior uveitis. There have been no large studies to determine the frequency of BSCR in the general population and its exact prevalence is currently unknown. BSCR accounts for 0.6–1.5% of patients with uveitis referred to tertiary centers for uveitis [7,8,9] and around 5–8% of patients with posterior uveitis [1]. The actual prevalence of BSCR in the general population is uncertain but likely to be around 0.1–0.6/100,000 [10].

## 3. Demographics

BSCR affects predominantly middle-aged patients. In one of the largest reviews on BSCR, Shah et al. [1] reported the mean age of disease onset to be 53 years with a range between 15 to 79 years. The disease only affects adult. Even though Shah et al. reported a slight feminine predominance of 54% [1], it may be explained by the increased life expectancy in women; there is likely no gender predominance in BSCR.

Patients with BSCR are predominantly white from European descent. BSCR in Latino-Hispanic, African-American, and Japanese individuals are limited to case reports [10].

## 4. Pathophysiology

The immune mechanisms involved in the pathogenesis of BSCR are not fully understood.

### 4.1. The Role of HLA-A29

HLA-A29 is involved in the presentation of peptide antigens to T cells. The association of HLA-A29 with BSCR was first reported 40 years ago [11]. More than 95% of BSCR patients carry the HLA-A29 allele. No other human disease is known have a stronger HLA allele association [4]. Since some rare BSCR cases have been reported in non-HLA-A29 carriers, its presence has not been asserted as a required criterion for the diagnosis (Table 1) [12]. Nevertheless, the association is so strong that other diagnoses should be considered in a HLA-A29 negative patient [4]. For some authors, the HLA-A29 is a “sine-qua non” criterion for the diagnosis [13]. They proposed to rename the disease as “HLA-A29 BSCR” [13].

However, the HLA-A29 allele is present in as many as 7–9% of Caucasians and most of these people do not develop BSCR; thus, the positive predictive value of HLA-A29 is weak [4]. HLA-A29 can be divided in at least 17 subtypes [15]. The HLA-A*29.01 and HLA-A*29.02 are the two main subtypes in the healthy population positive for HLA-29. The HLA-A*29.02 is strongly associated with BSCR, being observed in around 95% of patients [10]. It has been suggested that the HLA-A*29.02 subtype could confer risk for BSCR, whereas the HLA-A*29.01 could be protective, explaining the differences in prevalence between ethnicities. Indeed, the HLA-A*29.02 has a higher prevalence amongst Caucasians [16] and HLA-A*29.01 is predominant in Asians. However, although HLA-A*29.02 is predominant in African-Americans and Hispanics, they are protected from the disease [4]. These data show that the varying frequencies of HLA-A29 subtypes among different populations cannot explain the selectiveness of the disease for patients of European descent [17]. Moreover, a new HLA extended haplotype containing the A*2910 allele has also been shown to be associated with birdshot retinochoroidopathy and narrows susceptibility to the disease to the HLA molecule itself [17]. 

In addition, HLA-A*29:02 and HLA-A*29:01 differ by a single mutation (G376C/D102H), which does not seem to affect interaction with the presented peptides. 

Overall, the exact contribution of HLA-A29 to the pathophysiology of BSCR remains unclear. Indeed, HLA-A29 is relatively common in the European population (7–9%), and very few HLA-A29-positive individuals develop BSCR, suggesting that HLA-A29 is necessary but not sufficient to explain the pathophysiology of the disease, and other factors could be contributing to the development of BSCR [18].

### 4.2. The Role of the Endoplasmic Reticulum Aminopeptidase

A Genome Wide Association Study (GWAS) confirmed the association of HLA-A*29:02 with BCSR and found a new susceptibility locus: Endoplasmic reticulum aminopeptidase 2 (ERAP2) [19]. ERAP1 and ERAP2 are located in the endoplasmic reticulum (ER) and are involved in the final processing/presentation of antigenic precursors by HLA class I proteins [20]. ERAP1 and ERAP2 polymorphisms have also been reported in other HLA class I associated autoimmune diseases, such as ankylosing spondylitis, Crohn’s disease, and psoriasis [10].

Expression of ERAP1 and ERAP2 is coordinated, when ERAP1 expression decreases, ERAP2 expression increases [21]. Gelfman et al. suggested that increased ERAP2 along with decreased ERAP1 expression in BSCR cases would lead to higher availability of ERAP2-processed peptides for presentation onto HLA class I proteins [18]. Exceeding a peptide presentation threshold could activate the immune response in choroids of HLA-A29 carriers. However, this hypothesis does not provide an answer to the question of why BSCR is restricted to only ocular manifestations in distinction to other HLA-associated autoimmune disorders, which manifest more widely [17]. It has been speculated that an undefined antigen specifically and exclusively expressed in the choroid, retinal pigment epithelium, or perhaps outer retina is the pathogenic driver of the disease [17].

### 4.3. The Role of the T Cells

The strong association between HLA-A29 and the disease is suggestive of an autoreactive T-cell involvement in the pathogenesis of BSCR, although evidence of their direct implication is yet to be established. The disease is likewise believed to be T-cell driven because they are the dominant cells found on histopathology analyses [5,6]. Class I MHC molecules have an important role in presenting viral antigens to CD8+ T cells and there are numerous virus-specific CD8+ T cell epitopes identified that are presented in the context of HLA-A29 [15]. Autoimmune inflammation can develop from a presumed molecular mimicry initiated by infection [22]. Interestingly, analysis of vitreous fluid of two BSCR patients showed the presence of both CD4+ and CD8+ T cells [23]. More recently, Trombke et al. [24] analyzed the CD4+ and CD8+ T cell subsets in the peripheral blood of BSCR patients in relation to the activity of the disease. They observed a slight increase of effector memory CD8+ T cells expressing CD45RA in blood of inactive BSCR patients compared to the active group. Additionally, they identified a trend of decreased Th2 cells and increased Th1 cells in active BSCR [24]. 

Several studies suggest that T helper 17 (Th17) might be involved in the pathogenesis of BSCR. Th17 are a subset of CD4+ T helper cells that produce Interleukin (IL)-17 and play a role in various autoimmune diseases. Kuiper et al. showed increased levels of IL-17 in the aqueous humor of BSCR patients [25]. In addition, increased levels of cytokines associated with the differentiation of naïve T cells to Th17 (IL-23, IL1-beta, IL-6, and TGF-beta) have been shown in serum and ocular fluid from patients with BSCR [25,26]. In line with these results, Daien et al. showed an increase of Th17 cells in patients with BSCR [27]. In addition, an increased level of IL-17 in supernatant of PBMC (peripheral blood monocular cells) stimulated by retinal lysate has been reported in active BSCR patients compared to HLA-A29-positive controls [28].

The regulatory T cells (Treg cells) play a role in autoreactive T cells and antigen-presenting cells. Foster et al. showed no difference in the percentage of Treg cells between patients with BSCR (*n* = 5) compared to controls (*n* = 5), but they reported a decreased expression of Fox P3 in the BSCR group [29]. However, Daien et al. did not confirm these results, and rather found an increased level of Treg cells in 29 patients with BSCR compared to 16 controls [27].

## 5. Histopathology

To our knowledge, the histological findings of two cases of HLA-A29-positive BSCR have been reported [5,6]. They both described non-granulomatous lymphocytic infiltrates in the choroid and the ciliary body. Gaudio et al. described foci of lymphocytes in the deep choroid, the optic nerve head, and along the retinal vasculature [5]. The choroidal foci could correspond to the birdshot choroidal spots.

## 6. Clinical Presentation

The disease is bilateral, although cases of asymmetric involvement have been reported.

### 6.1. Clinical Symptoms

Visual symptoms are common in BSCR patients, even among those with good visual acuity [1,30]. Monnet et al. reported the visual complaints of 80 patients with BSCR analyzed with a standardized questionnaire [30]. The most common visual complaints were blurry vision, floaters, and nyctalopia reported in around two third of patients [30]. Abnormal contrast sensitivity and color vision were present in around 40% of cases [30]. In the review by Shah et al., 88% reported blurred vision, 43% floaters, 18% nyctalopia, and 9% dyschromatopsia [1]. Patients also complain of vibrating vision and loss of peripheral vision. Pain and photophobia are atypical and suggest an alternate diagnosis [10].

### 6.2. Clinical Signs

The diagnosis of BSCR is primarily based on the fundus examination and diagnostic criteria have been defined by Levinson et al. in 2006 [12] and more recently by the SUN (Standardization of Uveitis Nomenclature) group [14] (Table 1).

#### 6.2.1. Ocular Inflammation

Typically, the anterior segment has minimal inflammation. Synechia are never observed. There is no conjunctival injection. Laser flare photometry revealed no significant increase of aqueous humor protein concentration [1] and is, therefore, not recommended as a tool to monitor BSCR activity. The inflammatory vitreous reaction is mild without snowballs and shows variation in its intensity, tending to be more intense during early stages of the disease [30]. Optic disc edema and retinal vein vasculitis are often visible. The macular edema is the leading cause of vision loss in BSCR [30].

#### 6.2.2. Birdshot Lesions

Fundoscopy classically reveals multiple white-creamy choroidal ovoid lesions (Figure 1) that appear like a birdshot from a shotgun, explaining the name of this condition. The birdshot lesions are oval, measuring typically one-eighth to one-half optic disc diameter but they can be larger or confluent. They are usually clustered around the optic disc, most commonly nasal and inferior to the optic disc. In some cases, the lesions may develop hyperpigmentation (Figure 2). Peripapillary atrophy is a common finding observed during the follow-up and could be secondary to the coalescence of BSCR lesions surrounding the optic disc. In some cases, the birdshot lesions appeared many years after the first symptoms [31].

## 7. Functional Testing

### 7.1. Visual Acuity

The visual acuity is highly variable among BSCR patients ranging from “light perception” to 20/20. A visual acuity of 20/20 or better have been reported in 34% of eyes from 80 patients with BSCR [24]. In the review from Shah et al. assessing 213 patients from 34 papers, 62% of eyes had a visual acuity of 20/40 or better [1]. The visual acuity was fairly symmetrical with 75% of patients having not more than 2 Snellen lines different between eyes [1]. Legal blindness defined by a visual acuity of <20/200 has been reported in 5% to 14% of cases [30,32]. 

### 7.2. Visual Field Testing

Visual field abnormalities are common in patients with BCSR with different patterns including peripheral constriction, enlarged blind spot, and central/paracentral scotoma [1]. A correlation has been identified between visual acuity and visual field parameter, but the visual field can be abnormal even in patients with a visual acuity of 20/20 [33]. Thornes et al. showed that some visual field defects were reversed with the use of immunosuppressive drugs, which is suggestive that the visual field may be useful in monitoring the response to treatment [34]. Assessment of the peripheral visual field seems important; an abnormal central field was detected in around one-third of patients, whereas an abnormal peripheral field was identified in 75% of patients [34].

## 8. Multimodal Imaging

The clinical examination is key to making the diagnosis of BSCR but multimodal imaging including FA, ICGA, BAF, and OCT is useful in confirming the diagnosis and for monitoring disease activity. However, despite all these tools, monitoring the disease activity and progression remains challenging.

### 8.1. Fluorescein Angiography (FA)

Fluorescein angiography (FA) is a useful tool for monitoring retinal inflammation, more specifically, retinal vein vasculitis, which is characterized by vascular leakage and irregular venous caliber. Optic disc hyperfluorescence is a common finding (Figure 3 and Figure 4). BSCR patients frequently develop macular edema, characterized by a petaloid hyperfluorescent pattern. The choroidal neovascularization is a rare complication usually observed in later stages of the disease.

Birdshot lesions are described to be silent or to show early hypofluorescence and late mid/moderate hyperfluorescence [3,35]. However, due to their location inside the choroidal stroma, they are better visualized with fundus examination or indocyanine green angiography. 

Another finding first described by Gass is an increased arteriovenous transit time in BSCR [3]. Guex-Crosier et al. reported a mean arteriovenous transit time of 31 s in BSCR patients compared to 9 s in sarcoidosis and 7 s in VKH (Vogt Koyanagi Harada) patients [36]. This increased transit time in BSCR is supposed to be due to the extensive leakage of fluorescein into the retina from capillaries, delaying the filling of the veins [36]. Indeed, the arteriovenous time is not increased during ICG-A, likely because the ICG binds to larger proteins and does not leak from the retinal capillaries [37].

### 8.2. Indocyanine Angiography (ICG-A)

ICG-A is a useful tool for the diagnosis of BSCR when the birdshot lesions are not obvious on ophthalmoscopy [14] (Table 1 and Figure 4), for monitoring disease activity and assessing the effects of treatment [38].

Birdshot lesions appear hypofluorescent during the early and mid-phase of the ICG-A (Figure 4), which could be due to inflammatory infiltrates. During the late phase, many lesions became isofluorescent, which could indicate that the inflammatory lesions do not occupy the full thickness of the choroidal stroma [39]. On the other side, atrophic lesions remain hypofluorescent during late phase and colocalize with hyperfluorescent lesion on FA due to a window defect [39].

The number of birdshot lesions seen on ICG-A is usually greater than the number of lesions seen on fundus examination or fluorescein angiography [39]. They are typically round or oval in shape and located near a large choroidal vessel. 

The birdshot lesions can disappear during remission of the disease. Indeed, Cao et al. showed a decrease in the mean total area and number of lesions between the time of disease activity and disease quiescence [38].

### 8.3. Fundus Autofluorescence (FAF)

Blue fundus autofluorescence (FAF) identifies lipofuscin accumulation in the RPE and provides information about RPE/photoreceptor health. Hypoautofluorescent lesions are more common than hyperautofluorescent lesions in BSCR patients (Figure 2) [40]. The most common pattern is the peripapillary confluent hypoautofluorescence described in around 70% to 80% of BSCR patients [41,42,43,44]. Around half of the patients had hypoautofluorescent area in the macula, which has been associated with decreased visual acuity [41,42,43,44]. In a study evaluating 132 eyes, Bonï et al. identified an association between global hypoautofluorescence and disease duration or the presence of macular edema, suggesting that hypoautofluorescence could be a marker of BSCR severity [41]. Linear hypoautofluorescent streaks along the retinal vessels have been described in 28% to 50% of cases and are thought to represent retinal vasculitis [41,42]. 

No changes on FAF are observed at the level of the birdshot lesions except in cases where birdshot lesions are associated with RPE atrophy. Lesions in these cases are hypoautofluorescent (Figure 2).

A hyperautofluorescent area has also been reported in 14% to 40% of cases [41,44]. Hyperautofluorescence may indicate unmasking of underlying RPE autofluorescence by outer segments loss and may indicate an earlier form of visual dysfunction [41].

### 8.4. Optical Coherence Tomography (OCT)

The macular OCT is an excellent method to analyze the retinal and choroidal layers. It is widely used to detect and monitor macular edema (Figure 3), which will occur in the majority of patients during the follow-up of the disease [45]. OCT imaging also identified macular thinning and disruption of the photoreceptor ellipsoid zone, which have been associated with a decrease in visual acuity [45,46]. An increase in retinal nerve fiber layer (RNFL) thickness has been identified during the active phase of the disease [47]. A choroidal neovascularization is a rare complication with a hyperreflective fusiform aspect on OCT. More recently, an association has been shown between perivascular retinal thickness and the activity of retinal vasculitis evaluated on FA [48]. 

The use of the EDI (Enhanced Depth Imaging) technique or the swept source OCT allows for improved visualization of the choroid. Studies evaluating the choroid with EDI-OCT showed a generalized choroidal thinning (Figure 2), hyperreflective choroidal foci, increased choroidal reflectivity, and the presence of suprachoroidal hyporeflective spaces [49,50,51]. A correlation between the presence of subretinal fluid and photopsias, vasculitis, and vitreous haze has also been reported [52]. In a study evaluating 172 eyes with EDI-OCT, Boni et al. found areas of hyporeflectivity in the choroid in over 60% of eyes without relationship between these hyporeflective lesions and the clinical birdshot lesions [53]. Most of the studies assessing these choroidal morphological abnormalities had a cross-sectional design [51,53,54]. A longitudinal analysis of the choroidal thickness has been reported by Young et al. in a retrospective study evaluating 11 BSCR patients (22 eyes) with a median follow-up of 16 months. They showed a progressive choroidal thinning of 2.68 µm per month in BSCR eyes even in patients without active inflammation [55].

### 8.5. Optical Coherence Tomography Angiography (OCT-A)

Optical coherence tomography angiography (OCTA) is a new imaging modality that detects erythrocyte movement using the visualization of blood flow in vivo, without the use of intravenous dye, showing the retinal vasculature [56]. OCTA allows for segmentation of retinal layers and generates quantitative measurements, such as vessel densities (VD) and the foveal avascular zone (FAZ) area. Interestingly, Roberts et al. reported a decreased density in both superficial and deep plexus, in patients with BSCR, which correlated with visual acuity [57]. We confirmed these results in a larger cohort of 138 eyes (paper submitted). De Carlo et al. [58] identified qualitative abnormalities on OCT angiogram, such as capillary loops, focal dilatations, and an increased intercapillary space, which may correspond to capillary inflammation.

## 9. Other Tests

In addition to multimodal imaging and visual field, other tests have been used to monitor BSCR disease progression.

### 9.1. Color Vision and Contrast Sensitivity

An abnormal color vision on the desaturated L-15 test has been identified in 61% of patients with BSCR [59]. The color vision can be altered even in patients with a good visual acuity. Contrast sensitivity is decreased in almost all patients with BSCR, including those with 20/20 visual acuity and in absence of other ocular changes [60].

### 9.2. Electrophysiology

Electroretinogram (ERG) is widely used in many centers to monitor BSCR [10]. 

Abnormalities in ERG are common in patients with BSCR and have been reported in 72% to 89% of cases (Figure 5) [1].

In full-field ERG, the prolongation of the cone-mediated 30 Hz flicker time appears to be the most sensitive measure of damage by BSCR and can be improved by treatment [61]. This parameter has been used as an outcome measure in some clinical trials evaluating drug efficacy in BSCR [62,63].

An electronegative pattern has been described with a decrease in b-wave amplitude compared to a-wave amplitude, suggesting a dysfunction of the inner retina with relative sparing of the outer retina/inner choroidal complex [64,65]. This electronegative pattern reflects an alteration of the bipolar cells, the Müller cells, or the stimulus transmission from photoreceptor to bipolar cells with preserved photoreceptor function. This pattern does not appear in other etiologies of uveitis [10], but has been described in genetic diseases, such as juvenile X-linked retinoschisis, and acquired diseases, such as autoimmune retinopathy or retinal toxicity.

Photoreceptor dysfunction usually occurs later in the disease [66]. Rod dysfunction usually precedes cone dysfunction as the rod isolated b-wave is affected before the photopic b-wave and photopic flicker response in most patients. 

In the most advanced stage of the disease, the ERG is non-recordable; a situation that is essentially the same as retinitis pigmentosa [65]. 

Pattern ERG (PERG) reflects ganglion cell activity in the central retina. Few studies have evaluated this in BSCR [67]. In the study by Holder et al., PERG changes were similar to the full field ERG changes, in regards to amplitude decrease and peak time delay [61].

Multifocal ERG (mf ERG) measures the retinal function, especially in the central retina. Chiquet et al. showed that mfERG abnormalities were correlated with functional and anatomical parameters [68]. The same group reported a deterioration of the mfERG during a 5-year follow-up, where the visual acuity and visual field remained unchanged, suggesting a higher sensitivity of the mfERG in monitoring disease progression [69].

## 10. Differential Diagnosis

The recently published classification criteria (Table 1) developed by the SUN working group for BSCR has a low misclassification rate, therefore indicating good diagnosis performance [14]. BSCR can usually be distinguished from other diseases by history, clinical examination, multimodal imaging, and the HLA-A29 test. However, syphilis, tuberculosis, sarcoidosis, multifocal choroiditis, and ocular lymphoma should be considered in the differential diagnosis. The following tests can rule out the main differential diagnosis: syphilis serology, ACE level, Interferon gamma release assay, and chest X-ray [10]. For some investigators, all cases diagnosed as birdshot in HLA A29 negative patients may in fact have one of these differential diagnoses, in particular sarcoidosis [13,14].

## 11. Clinical Course and Prognosis

BSCR is a chronic progressive disease, but the range of BSCR phenotypes varies between a form of the disease with good prognosis to sight-threatening, progressive chorioretinal atrophy. Little is known about the long-term course of untreated BSCR patients. However, of 158 patients from our cohort, 6% did not receive any treatment and maintained a good visual acuity of 20/20 with a normal visual field after a mean follow-up of 10 years (unpublished data). These results suggest that a subgroup of patients had a “benign” form of the disease, as previously described by Lages V et al. [70].

Nevertheless, most patients have multiple inflammatory exacerbations with progressive visual loss. Macular edema is the most common cause of vision loss and will occur in most patients over the course of the disease. Chronic macular edema can lead to photoreceptor dysfunction and atrophy. Choroidal neovascularization is a rare complication of BSCR. Touhami et al. evaluated the prognosis factors for long term visual outcomes in 56 patients with BSCR. They showed that patients with overt inflammatory signs, such as vitritis, macular edema, and choroidal spots on ICG-A had better visual outcomes. These intriguing results could be explained by the fact that these patients were treated by corticosteroids earlier than patients with less prominent inflammatory signs [71]. 

## 12. Treatment

Currently, there is no consensus for treatment initiation, optimal regime, or duration. It is usually recommended to treat BSCR patients with corticosteroids associated with immunosuppressive or biological therapy. Although there is no international consensus, we propose a treatment algorithm for the management of BSCR based on a literature review and our experience (Figure 6). Indeed, many different drugs have been used and there are no randomized controlled trials focused on BSCR to prove the efficacy of these treatments. 

### 12.1. Corticosteroids

Systemic corticosteroids are commonly used in the management of acute inflammatory manifestations of the disease. However, their efficacy is limited for long-term control of the disease at a low dose, and their side effects prevent their long-term use at a high dose [72]. They are usually associated with an immunosuppressive or biological drug, allowing the tapering of oral corticosteroid to a level which is safe for long-term use (<7.5 mg/d).

Sub-tenon triamcinolone acetonide or intravitreal dexamethasone implants (Ozurdex, Allergan) are commonly used in the treatment of macular edema with an increased risk of cataract and glaucoma [10]. Intravitreal fluocinolone acetonide implant has demonstrated efficacy in treating the inflammatory signs of BSCR, but were associated with cataract progression and glaucoma [73]. In a retrospective case series of 11 BSCR patients, Ajamil-Rodanes et al. showed the efficacy of fluocinonole in treating vasculitis and macula edema, whereas no effect was observed on the choroidal lesions that persisted despite the treatment [74]. 

### 12.2. Immunosuppressive Drugs

Early treatment with steroid sparing immunosuppressive/immunomodulatory therapy (IMT) can limit steroid side effects and preserve visual function [72]. You et al. reported the outcomes of 132 patients treated by IMT showing a decreased rate of BSCR exacerbation after 5 years of treatment [75]. Early and sufficiently dosed immunosuppressive treatment could event prevent the appearance of birdshot lesions [76]. Thorne et al. showed that the use of immunosuppressive drugs reduced the risk of developing macular edema, whereas oral corticosteroid therapy at a daily dose of less than 15 mg did not [77].

The immunosuppressive drugs can be used alone or in combination in refractory cases [10].

Ciclosporin has been used, as it inhibits T lymphocytes, which could play a central role in BSCR pathogenesis. Ciclosporin is an effective treatment to maintain visual acuity and limit the disease progression [10,72]. However, the use of cyclosporin is limited by its side effects of nephrotoxicity and hypertension. Tacrolimus is another immunosuppressive drug inhibiting the T-cell proliferation. The safety profile of tacrolimus is better than ciclosporin in term of renal toxicity and risk of hypertension. In a retrospective case series of 25 BSCR patients, Islam et al. reported a good safety profile in 84% of patients, associated with an improvement of both visual and functional parameters [78].

Antimetabolites agents such as azathioprine, methotrexate, or mycophenolate mofetil (MMF) have been used as steroid sparing agents with favorable results. MMF is a widely used drug, which has been proven effective and well tolerated [79]. Kiss et al. reported a durable remission of patients treated with cyclosporine and MMF. The treatment was maintained until the disease had remained quiescent for 2 years and slowly tapered [72]. 

### 12.3. Biotherapies

Biologic agents have been used successfully to treat BSCR. The TNF alpha inhibitors infliximab, a monoclonal chimeric antibody, and more recently adalimumab, a humanized anti-TNF alpha antibody have been used for the treatment of BSCR patients refractory to conventional immunosuppressive therapy. Adalimumab obtained FDA approval for intermediate and posterior non-infectious uveitis based on the results of the VISUAL trial, a multicenter randomized phase 3 study in which 30 out of the 229 patients included had a birdshot chorioretinopathy [80]. Adalimumab has also been shown to be effective in improving the visual acuity in a report of 19 patients with BSCR refractory to immunosuppressive drugs, but complete remission was rarely achieved [81].

Daclizumab, a monoclonal antibody against the IL-2 receptor of T cells, was effective in decreasing inflammation in 8 patients with BSCR whose disease was either refractory or were intolerant to immunosuppressive therapy [82]. 

Evidence supporting the use of anti-IL6 receptor, tocilizumab, is limited to a case series showing inflammation being controlled by BSCR refractory to anti-TNF-alpha treatment [83,84].

### 12.4. Intravenous Immunoglobulin (IV Ig)

LeHoang et al. reported the outcomes of 18 patients treated with IV Ig, showing an improvement of visual acuity and macular edema in around half of the eyes [85].

## 13. Conclusions

Birdshot chorioretinopathy (BSCR) is presumably an auto-immune disease of the eye affecting middle-aged individuals from European descent and is strongly associated with the HLA-A29. The immune mechanisms involved in the pathogenesis of BSCR are not fully understood. The clinical examination is key to making the diagnosis of BSCR, but multimodal imaging including FA, ICGA, FAF, and OCT is useful in confirming the diagnosis and monitoring disease activity. Corticosteroids at low doses associated with immunosuppressive or biological therapies are used to treat the disease, although there is no consensus for the optimal treatment regime and duration.

## Figures and Tables

**Figure 1 jcm-11-04772-f001:**
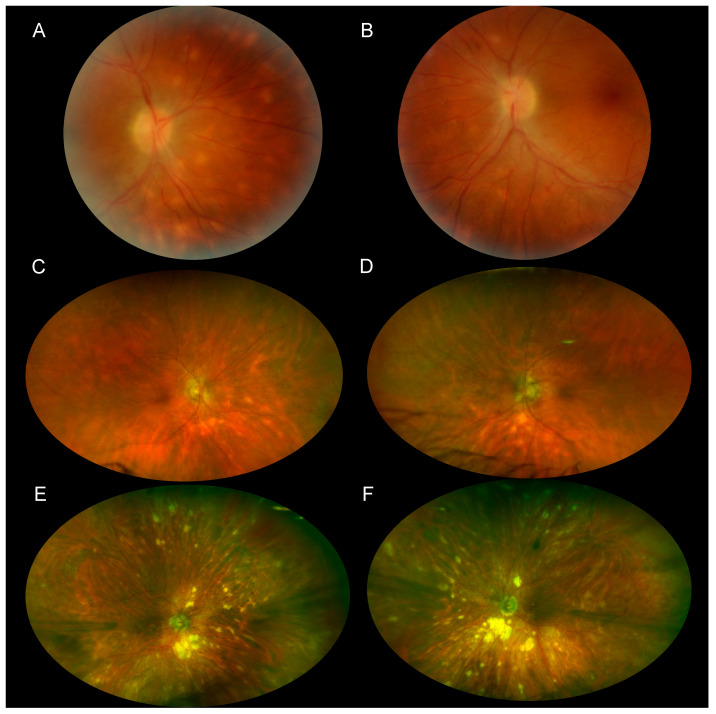
(**A**,**B**) Color fundus photography of a 38-year-old man with recently diagnosed BSCR. The round to oval birdshot lesions are concentrated inferonasally to the optic disc. (**C**,**D**) Wide-field color fundus photography of an 80-year-old man taken 20 years after the diagnosis of BSCR. The birdshot lesions appear as oval or linear streaks of hypopigmentation and remain more prominent inferonasally to the optic disc. (**E**,**F**) Wide-field color fundus photography of a 95-year-old woman taken 18 years after the diagnosis of BSCR. Birdshot lesions are ovoid in shape and appear to radiate from the optic disc. Some lesions coalesce into areas of chorioretinal atrophy predominantly nasally from the optic disc.

**Figure 2 jcm-11-04772-f002:**
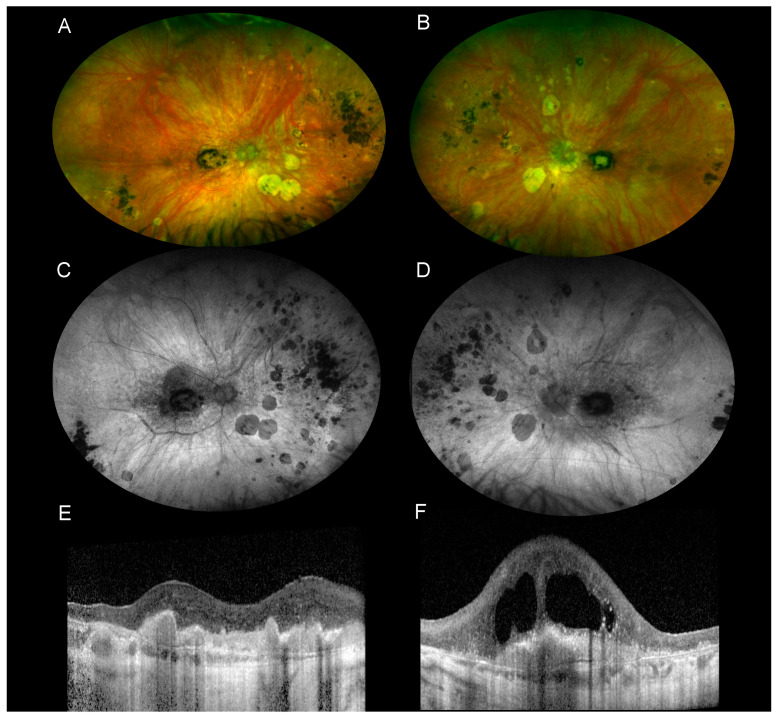
Multimodal imaging of a 91-year-old woman with long-lasting BSCR complicated with macular neovascularization. The BSCR was diagnosed 30 years ago. (**A**,**B**) Wide-field color fundus photography showed birdshot lesions with chorioretinal atrophy predominantly nasally from the optic disc. Pigmented changes are seen at the level of birdshot lesions and at the macula. (**C**,**D**) Wide-field fundus autofluorescence shows confluent hypoautofluorescence of the birdshot lesions due to RPE atrophy of the macula and around the optic disc in both eyes. (**E**,**F**) OCT B scans identify hyperreflective fusiform lesions at the level of the macula in both eyes, consistent with a fibrovascular scar. There is no subretinal or intraretinal fluid in the right eye, whereas there is degenerative macular edema in the left eye. The choroid is very thin in both eyes.

**Figure 3 jcm-11-04772-f003:**
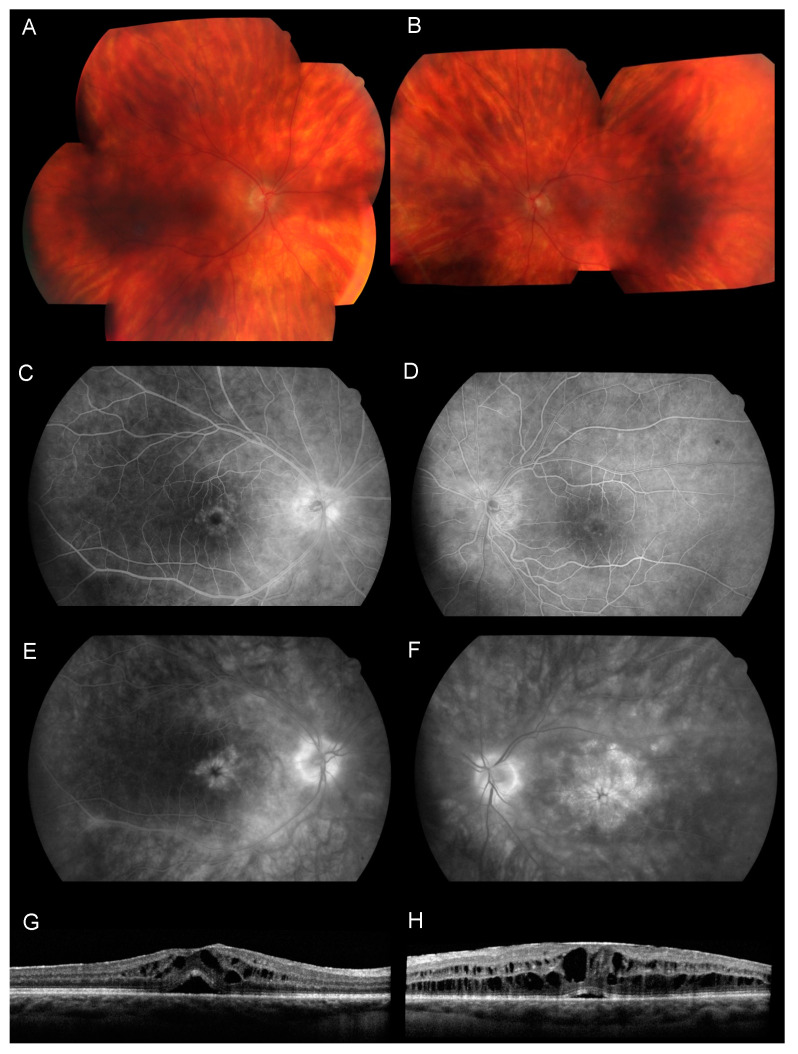
Multimodal imaging of a 46-year-old woman with recently diagnosed BCR. (**A**,**B**) Color fundus photography shows oval to linear creamy birdshot lesions. (**C**–**F**) Early-phase (**C**,**D**) and late-phase (**E**,**F**) fluorescein angiography shows cystoid macular edema associated with hyperfluorescence of the optic disc in both eyes. (**G**,**H**) OCT B-scan shows cystoid macular edema with subretinal fluid in both eyes.

**Figure 4 jcm-11-04772-f004:**
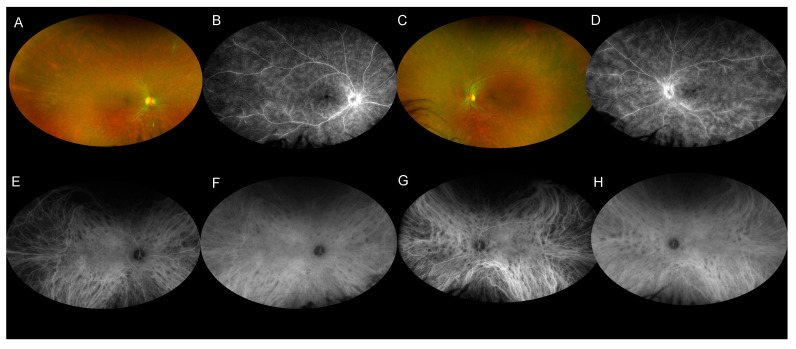
Multimodal imaging of a 30-year-old man with a recently diagnosed BSCR. (**A**,**C**) Wide-field color fundus photography shows subtle birdshot lesions inferonasally to the optic disc. (**B**,**D**) Late-phase wide-field fluorescein angiography illustrates hyperfuorescence of the optic disc and retinal veins in both eyes. (**E**–**H**) Wide-field indocyanine angiography (ICG-A) shows hypofluorescent spots consistent with birdshot lesions at the early-phase (**E**,**G**) and the mid-phase. The birdshot lesions are better seen on ICGA than on color fundus photography in this patient.

**Figure 5 jcm-11-04772-f005:**
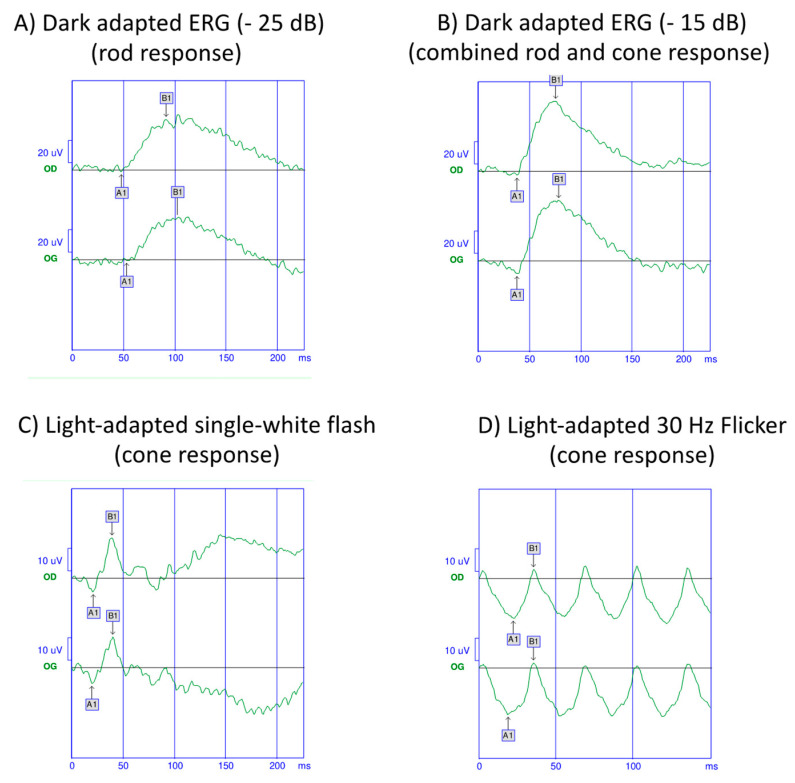
Global electroretinogram (ERG) of a 55-year-old man with birdshot chorioretinopathy. The rod-isolated ERG response (**A**) and combined rod cone response (**B**) are abnormal with a reduced amplitude of the a and b waves in both eyes. (**C**,**D**) The cones mediated responses are abnormal with a reduced amplitude and an increased delay.

**Figure 6 jcm-11-04772-f006:**
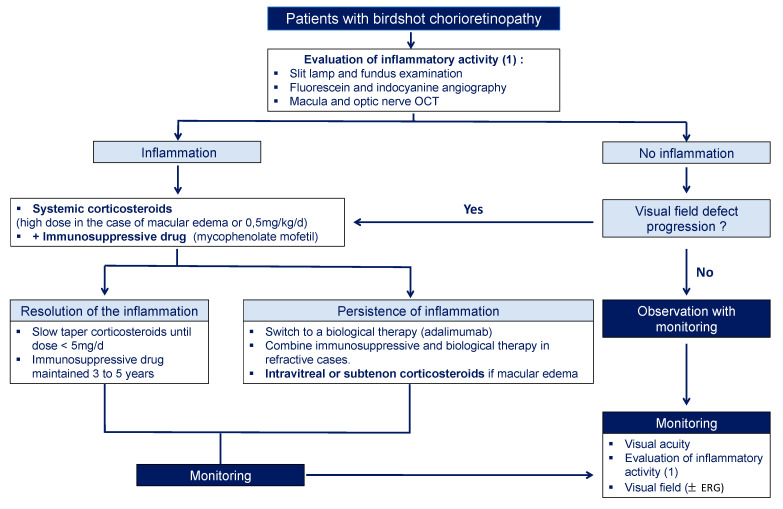
Treatment and monitoring algorithm proposed by the authors based on the literature review and the authors’ experience.

**Table 1 jcm-11-04772-t001:** Birdshot Chorioretinopathy Diagnostic Criteria (adapted from the Standardization of Uveitis Nomenclature (SUN) working group [14]). Birdshot Chorioretinopathy Diagnostic = Criteria 1, 2 and 3 or 4.

	Characteristic bilateral multifocal choroiditis on ophthalmoscopy: (a) multifocal cream-colored or yellow-orange, oval or round choroidal lesions (“birdshot spots”)
AND	2. Absent to mild anterior chamber inflammation (a) Absent to mid anterior chamber cells AND (b) No keratic precipitates AND (c) No posterior synechiae
AND	3. Absent to moderate vitritis
OR	4. Multifocal choroiditis with (a) Positive HLA-A29 test AND either (b or c) (b) Characteristic “birdshot” spots on ophthalmoscopy (c) Characteristic indocyanine green angiogram (multifocal hypofluorescent spots) without characteristic “birdshot” spots on opthalmoscopy
Exclusions	Positive serologic test for syphilis using a treponemal test
	2. Evidence of sarcoidosis (either bilateral hilar adenopathy on chest imaging or tissue biopsy demonstrating non caseating granulomata)
	3. Evidence of intraocular lymphoma on diagnostic vitrectomy or tissue biopsy

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
