# Peer review of "Birdshot Chorioretinopathy: A Review"

_jcm, 2022, doi:10.3390/jcm11164772_

Round 1
Reviewer 1 Report
Well-structured bibliographic review, although most of the bibliographic citations are not current.The new treatments are not explained in detail
Author Response
We thank the reviewer for his positive comments.
We have added a paragraph on the biological therapies and intravenous immunoglobulin in the revised version of the manuscript as follow (line 437-449):
“The TNF alpha inhibitors, infliximab, a monoclonal chimeric antibody and more recently adalimumab, a humanized anti-TNF alpha antibody have been used for the treatment of BSCR patients refractory to conventional immunosuppressive therapy. Adalimumab obtained FDA approval for intermediate and posterior non-infectious uveitis based on results of the VISUAL trial, a multicenter randomized phase 3 study in which 30 out of the 229 patients included had a birdshot chorioretinopathy.[75] Adalimumab has also been shown to be effective in improving the visual acuity in a report of 19 patients with BSCR refractory to immunosuppressive drugs, but complete remission was rarely achieved. [76]
Daclizumab, a monoclonal antibody against the IL-2 receptor of T cells was effective in decreasing inflammation in 8 patients with BSCR whose disease was either refractory or were intolerant to immunosuppressive therapy.[77]
Additionally, we have added a paragraph on the intravenous immunoglobulin treatment (line 454-457):
“LeHoang et al reported the outcomes of 18 patients treated with IV Ig showing an improvement of visual acuity and macular edema in around half of the eyes.[80]”

Reviewer 2 Report
Elodie Bousque et al reported a review on Birdshot chorioretinopathy (BSCR), which is a rare inflammatory eye disease. In the current version, although this article has sorted out some of the research progress on BSCR disease, it has not yet put forward very valuable new ideas on this topic compared with previous studies, such as the review by Alastair K. Denniston et al. (PMID:27175923). T cell and immune imbalance in BSCR may be a very important feature, which has not been well reviewed in this article. If this article only focuses on clinical diagnosis, as stated in line 42, other parts of the article are unnecessary, and the title should be adjusted accordingly.
Minor concerns:
1. The number of keywords on line 22 does not reach three or more.
2. All the images used need to be licensed for copyright.
3. Format errors in lines 125, 314, 364, and 400.
4. et al. should be italicized in the whole text.
Author Response
Reviewer 2 :
Elodie Bousque et al reported a review on Birdshot chorioretinopathy (BSCR), which is a rare inflammatory eye disease. In the current version, although this article has sorted out some of the research progress on BSCR disease, it has not yet put forward very valuable new ideas on this topic compared with previous studies, such as the review by Alastair K. Denniston et al. (PMID:27175923). T cell and immune imbalance in BSCR may be a very important feature, which has not been well reviewed in this article. If this article only focuses on clinical diagnosis, as stated in line 42, other parts of the article are unnecessary, and the title should be adjusted accordingly.
We have added a paragraph on the T cell involvement in the revised version of the manuscript (line 115 to line 145):
“The strong association between HLA-A29 and the disease is suggestive of an auto-reactive T-cell involvement in the pathogenesis of BSCR, although evidence of their direct implication is yet to be established. The disease is likewise believed to be T-cell driven because they are the dominant cells found on histopathology analyses.[5,6] Class I MHC molecules have an important role in presenting viral antigens to CD8+ T cells and there are numerous virus-specific CD8+T cell epitopes identified that are presented in the context of HLA-A29.[14] Autoimmune inflammation can develop from a presumed molecular mimicry initiated by infection.[21] Interestingly, analysis of vitreous fluid of two BSCR patients showed the presence of both CD4+ and CD8+ T cells.[22] More recently, Trombke J et al [23] analyzed the CD4+ and CD8+ T cell subsets in the peripheral blood of BSCR patients in relation to the activity of the disease. They observed a slight increase of effector memory CD8+Tcells expressing CD45RA in blood of inactive BSCR patients compared to the active group. Additionally, they identified a trend of decreased Th2 cells and increased Th1 cells in active BSCR.[23]
Several studies suggest that T helper 17 (Th17) might be involved in the patho-genesis of BSCR. Th17 are a subset of CD4+ T helper cells that produce Interleukin (IL)-17 and play a role in various autoimmune diseases. Kuiper J et al showed increased levels of IL-17 in the aqueous humor of BSCR patients.[24] In addition, increased levels of cytokines associated with the differentiation of naïve T cells to Th17 (IL-23, IL1-beta, IL-6 and TGF-beta) have been shown in serum and ocular fluid from patients with BSCR.[24,25] In line with these results, Daien V et al showed an increase of Th17 cells in patients with BSCR.[26] In addition, an increased level of IL-17 in supernatant of PBMC (peripheral blood monocular cells) stimulated by retinal lysate has been reported in active BSCR patients compared to HLA-A29 positive controls.[27]
The regulatory T cells (Treg cells) play a role in autoreactive T cells and antigen presenting cells. Foster et al showed no difference in the percentage of Treg cells between patients with BSCR (n=5) compared to controls (n=5), but they reported a decreased ex-pression of Fox P3 in the BSCR group.[28] However, Daien et al did not confirm these results, and rather found an increased level of Treg cells in 29 patients with BSCR compared to 16 controls.[26]”
Minor concerns:
- The number of keywords on line 22 does not reach three or more.
We have added keywords and there are now 5 keywords.
- All the images used need to be licensed for copyright.
Images are from patients followed at Cochin Hospital - Université Paris. The patients belong to the CO-BIRD cohort ( ClinicalTrials.gov Identifier: NCT05153057). They have signed an informed consent which allows the use of their anonymized medical data.
- Format errors in lines 125, 314, 364, and 400.
We have not been able to detect the errors. Please specify, if you observe the errors again.
- et al. should be italicized in the whole text.
Done

Reviewer 3 Report
The authors have conducted a very well organized review. The following are the points need to be addressed more:
1 More details on the electrophysiologic findings compared to other differentials. Also adding an ERG image will be very helpful for better understanding of the condition.
2 More details on the treatment options to provide a good practical reference for clinicians.
Author Response
Reviewer 3 :
The authors have conducted a very well organized review. The following are the points need to be addressed more:
We thank the reviewer for his encouraging comments.
1 More details on the electrophysiologic findings compared to other differentials. Also adding an ERG image will be very helpful for better understanding of the condition.
We have added the following information in the ERG paragraph (line 340- line 358) and we have also added a figure with a global ERG (Figure 5).
“This parameter has been used as an outcome measure in some clinical trials evaluating drug efficacy in BSCR.[59], [60]
An electronegative pattern has been described with a decrease in b-wave amplitude compared to a-wave amplitude suggesting a dysfunction of the inner retina with relative sparing of outer retina/inner choroidal complex.[61,62] This electronegative pattern reflects an alteration of the bipolar cells, the Müller cells or of the transmission of the stimulus from photoreceptor to bipolar cells with preserved photoreceptor function. This pattern does not appear in other etiologies of uveitis[10] but has been described in genetic diseases such as juvenile X-linked retinoschisis, in acquired diseases such as autoimmune retinopathy or retinal toxicity.
Pattern ERG (PERG) reflects ganglion cells activity in the central retina. Few studies have evaluated in BSCR.[64] In Holder et al study, PERG changes were similar to the full field ERG changes in term of amplitude decrease and peak time delay.[58]”
2 More details on the treatment options to provide a good practical reference for clinicians.
We have added a paragraph on biological therapies and proposed an algorithm of treatment in the revised manuscript (Figure 6).

Reviewer 4 Report
This is a manuscript that describes Birshot's choriretinopathy in detail.
It is well-written and covers everything from epidemiology and pathophysiology to treatment and prognosis. The different areas are described in adequate detail and depth.
Only a few notes are addressed to the authors.
. Line 338- For some investigators, all cases diagnosed as birdshot in HLA A29 negative patients may in fact have one of these differential diagnoses, in particular sarcoidosis. Please show reffrences
. Fig 3- Please consider to show show early / mid phase fluorescein angiography
. Conclusions: Birdshot chorioretinopathy (BSCR) is presumably an auto-immune disease of the eye affecting middle-aged individuals of European descent and strongly associated with the HLA-A29.
Please consider to replace by:
Birdshot chorioretinopathy (BSCR) is presumably an auto-immune disease of the eye affecting predominantly white middle-aged individuals from European descent and strongly associated with the HLA-A29.
Author Response
This is a manuscript that describes Birshot's choriretinopathy in detail.
It is well-written and covers everything from epidemiology and pathophysiology to treatment and prognosis. The different areas are described in adequate detail and depth.
We thank the reviewer for his positive comments.
Only a few notes are addressed to the authors.
. Line 338- For some investigators, all cases diagnosed as birdshot in HLA A29 negative patients may in fact have one of these differential diagnoses, in particular sarcoidosis. Please show reffrences
We have now added 2 papers, which underline the importance of HLA-A29 (line 381):
13) Standardization of Uveitis Nomenclature (SUN) Working Group Classification Criteria for Birdshot Chorioretinitis. Am J Ophthalmol 2021
14) Papadia, M.; Pavésio, C.; Fardeau, C.; Neri, P.; Kestelyn, P.G.; Papasavvas, I.; Herbort, C.P. HLA-A29 Birdshot Retinochoroiditis in Its 5th Decade: Selected Glimpses into the Intellectual Meanderings and Progresses in the Knowledge of a Long-Time Misunderstood Disease. Diagnostics (Basel) 2021,
. Fig 3- Please consider to show show early / mid phase fluorescein angiography
We have now added the early-phase of the fluorescein angiography in the Figure 3 (line 253).
. Conclusions: Birdshot chorioretinopathy (BSCR) is presumably an auto-immune disease of the eye affecting middle-aged individuals of European descent and strongly associated with the HLA-A29.
Please consider to replace by:
Birdshot chorioretinopathy (BSCR) is presumably an auto-immune disease of the eye affecting predominantly white middle-aged individuals from European descent and strongly associated with the HLA-A29.
Done

Reviewer 5 Report
I would like to thank the authors for the opportunity to read this article.
The authors present a complete review of Birdshot chorioretinitis. The article is well written and provides important information about diagnosis and management of the disease. I have some minor points that I believe the authors should include.
Minor reviews
1. The authors have mentioned the possible agents that can be used to treat this disease but I believe they should mention Tacrolimus which is a safer and better solution than cyclosporine. A good reference can be the article : Islam F, Westcott M, Rees A, Robson AG, Kapoor B, Holder G, Pavesio C. Safety profile and efficacy of tacrolimus in the treatment of birdshot retinochoroiditis: a retrospective case series review. Br J Ophthalmol. 2018 Jul;102(7):983-990. doi: 10.1136/bjophthalmol-2017-310436. Epub 2017 Oct 19. PMID: 29051329.
2. The authors mention the diagnostic criteria provided by SUN but there are some well-known specialists who are published a different point of view; they proposed slightly different diagnostic criteria underlying the importance of the HLA-A29 in the diagnosis of the Birdshot chorioretinopathy. As regards my opinion, the authors should mention this different point of view ( article : Papadia M, Pavésio C, Fardeau C, Neri P, Kestelyn PG, Papasavvas I, Herbort CP. HLA-A29 Birdshot Retinochoroiditis in Its 5th Decade: Selected Glimpses into the Intellectual Meanderings and Progresses in the Knowledge of a Long-Time Misunderstood Disease. Diagnostics (Basel). 2021 Jul 19;11(7):1291. doi: 10.3390/diagnostics11071291. PMID: 34359373; PMCID: PMC8305470.)
Author Response
The authors present a complete review of Birdshot chorioretinitis. The article is well written and provides important information about diagnosis and management of the disease. I have some minor points that I believe the authors should include.
We thank the reviewer for his comments and the opportunity to improve our review.
Minor reviews
- The authors have mentioned the possible agents that can be used to treat this disease but I believe they should mention Tacrolimus which is a safer and better solution than cyclosporine. A good reference can be the article : Islam F, Westcott M, Rees A, Robson AG, Kapoor B, Holder G, Pavesio C. Safety profile and efficacy of tacrolimus in the treatment of birdshot retinochoroiditis: a retrospective case series review. Br J Ophthalmol. 2018 Jul;102(7):983-990. doi: 10.1136/bjophthalmol-2017-310436. Epub 2017 Oct 19. PMID: 29051329.
We have added the following paragraph with the citation mentioned by the reviewer in the revised review (line 436) :
“Tacrolimus is another immunosuppressive drug inhibiting the T-cell proliferation. The safety profile of tacrolimus is better than ciclosporin in term of renal toxicity and risk of hypertension. In a retrospective case series of 25 BSCR patients, Islam et al reported a good safety profile in 84% of patients associated with an improvement of both visual and functional parameters.[76] »
- The authors mention the diagnostic criteria provided by SUN but there are some well-known specialists who are published a different point of view; they proposed slightly different diagnostic criteria underlying the importance of the HLA-A29 in the diagnosis of the Birdshot chorioretinopathy. As regards my opinion, the authors should mention this different point of view ( article : Papadia M, Pavésio C, Fardeau C, Neri P, Kestelyn PG, Papasavvas I, Herbort CP. HLA-A29 Birdshot Retinochoroiditis in Its 5th Decade: Selected Glimpses into the Intellectual Meanderings and Progresses in the Knowledge of a Long-Time Misunderstood Disease. Diagnostics (Basel). 2021 Jul 19;11(7):1291. doi: 10.3390/diagnostics11071291. PMID: 34359373; PMCID: PMC8305470.)
We have added the reference with the following sentence as proposed by the reviewer (line 70):
“For some authors, the HLA-A29 is a “sine-qua non” criterion for the diagnosis. [14] They proposed to rename the disease as “HLA-A29 BSCR”.[14]”

Reviewer 6 Report
This is a very complete review of Birdshot chorioretinopathy; I have just two comments:
-Regarding choroidal OCT: the authors comment on several findings like thickness, reflectivity and hyperreflective foci; It would be interesting to specify what happens with this parameters during the evolution of the disease.
-Regarding the fluocinolone implant: the authors state that it has "efficacy in treating the inflammatory signs..."; it should be specified whether the response of each of the inflammatory components (vasculitis, choroidal foci, macular edema, vitritis) is the same with this treatment.
Author Response
This is a very complete review of Birdshot chorioretinopathy; I have just two comments:
Thank you for your comments, which allow us to improve our review.
-Regarding choroidal OCT: the authors comment on several findings like thickness, reflectivity and hyperreflective foci; It would be interesting to specify what happens with this parameters during the evolution of the disease.
Unfortunately, most of the studies assessing the choroid are cross-sectional and the evolution of the reflectivity or hyperreflective foci is unknown.
To answer the reviewer question, we have now added the study of Young et al, Retina 2015 evaluating the changes of the choroidal thickness in BSCR as stated below (line 318):
“Most of the studies assessing these choroidal morphological abnormalities had a cross-sectional design.[51,53,54] A longitudinal analysis of the choroidal thickness has been reported by Young et al in a retrospective study evaluating 11 BSCR patients (22 eyes) with a median follow-up of 16 months. They showed a progressive choroidal thinning of 2.68 µm per month in BSCR eyes even in patients without active inflammation.[55]”
-Regarding the fluocinolone implant: the authors state that it has "efficacy in treating the inflammatory signs..."; it should be specified whether the response of each of the inflammatory components (vasculitis, choroidal foci, macular edema, vitritis) is the same with this treatment.
We have now added the following sentence (line 423):
“In a retrospective case series of 11 BSCR patients treated with fluocinolone, Ajamil-Rodanes et al showed the efficacy of fluocinonole in treating vasculitis and macula edema whereas no effect was observed on the choroidal lesions that persisted despite the treatment.”